# Engineered Cu-PEN Composites at the Nanoscale: Preparation and Characterisation

**DOI:** 10.3390/nano12071220

**Published:** 2022-04-05

**Authors:** Jana Pryjmaková, Mariia Hryhoruk, Martin Veselý, Petr Slepička, Václav Švorčík, Jakub Siegel

**Affiliations:** 1Department of Solid State Engineering, University of Chemistry and Technology Prague, 166 28 Prague, Czech Republic; hryhorum@vscht.cz (M.H.); petr.slepicka@vscht.cz (P.S.); vaclav.svorcik@vscht.cz (V.Š.); 2Department of Organic Technology, University of Chemistry and Technology Prague, 166 28 Prague, Czech Republic; veselyr@vscht.cz

**Keywords:** laser treatment, nanostructured polymer, periodic structures, nanowires, nanocomposites

## Abstract

As polymeric materials are already used in many industries, the range of their applications is constantly expanding. Therefore, their preparation procedures and the resulting properties require considerable attention. In this work, we designed the surface of polyethylene naphthalate (PEN) introducing copper nanowires. The surface of PEN was transformed into coherent ripple patterns by treatment with a KrF excimer laser. Then, Cu deposition onto nanostructured surfaces by a vacuum evaporation technique was accomplished, giving rise to nanowires. The morphology of the prepared structures was investigated by atomic force microscopy and scanning electron microscopy. Energy dispersive spectroscopy and X-ray photoelectron spectroscopy revealed the distribution of Cu in the nanowires and their gradual oxidation. The optical properties of the Cu nanowires were measured by UV-Vis spectroscopy. The sessile drop method revealed the hydrophobic character of the Cu/PEN surface, which is important for further studies of biological responses. Our study suggests that a combination of laser surface texturing and vacuum evaporation can be an effective and simple method for the preparation of a Cu/polymer nanocomposite with potential exploitation in bioapplications; however, it should be borne in mind that significant post-deposition oxidation of the Cu nanowire occurs, which may open up new strategies for further biological applications.

## 1. Introduction

Due to the development of new technologies and techniques over the last few decades, a growing emphasis has been placed on the useful properties of already-known materials. Common materials such as polymers have become an integral part of our lives and can be used in many applications, from simple packages to sophisticated devices. However, some applications require enhanced properties because polymers are usually predisposed to degradation and fragility. For this reason, researchers worldwide focus on the design, chemical composition, and properties of the final product, which can be improved by using nanostructured materials. Prospective engineered nanomaterials have great potential in many areas such as electronic [1], textile [2], petrochemical [3], and pharmaceutical [4] industries, as well as in biology [5,6], agriculture [7], or environmental [8] applications. In addition to the applications of nanomaterials themselves, they can also be used to create composites with irreplaceable properties.

Nanomaterials, together with tools of surface modification, represent promising approaches for ensuring specific chemical and physical properties that can extend the range of applicability of polymer materials. There are a wide variety of surface modification techniques, such as physical and chemical deposition of the thin layer [9,10], ion implantation [11], Langmuir-Blodgett coating [12], or plasma [13] and laser treatment [14,15], which influence surface topography, chemistry, and wettability. Laser treatment is a very effective method to modify the surface of polymers [16,17]. The interaction of the laser beam with the polymer can provide nanopatterning in the form of dimples or ripples known as laser-induced periodic surface structures (LIPSS) [18]. One of the advantages of laser treatment is the possibility to create various forms of nanostructures by changing the laser fluence and surrounding environment. In that case, the surface of the polymer can be formed into worm-like or globular-like structures [19].

The creation of LIPSS offers a basis for thin-film deposition with considerable surface area [20,21] and specific structures (nanowires) [22,23,24]. Notably, metallic materials are attractive as coatings and reinforcements for polymeric composites [25,26]. One of the promising materials is copper, which is known to be an excellent electrical and thermal conductor and an effective antibacterial agent (concurrently with silver, platinum, or palladium). The advantages of copper-based nanocomposites include the change in wettability and mechanical properties [27,28]. For this reason, nanostructured copper can be used, e.g., in food packaging [29,30], electronic packaging [31,32], telecommunications [33], solar cells [34], or some medical treatments and devices [35,36,37]. Even complex devices such as pressure-responsive sensors based on copper nanowires combined with graphene oxide can be engineered [38].

In this study, we introduce an advanced composite nanomaterial with enhanced surface properties. We treated the polyethylene naphthalate (PEN) with a 248-nm KrF excimer laser combined with vacuum evaporation of Cu, which resulted in coherent metal nanowires being evenly distributed over the polymer surface. Atomic force microscopy (AFM) was used to investigate the change in surface morphology, which is important for nanowire formation and interaction of the final material with pathogens or, equally, with healthy cells. The chemical composition and distribution of the separated Cu nanowires were analysed by angle-resolved X-ray photoelectron spectroscopy (AR-XPS) and energy-dispersed X-ray spectroscopy (EDS). The optical properties of the prepared composites were determined by UV-Vis spectroscopy. Following our intention to study the biological activity of such nanocomposites in future experiments, their wettability was also investigated.

## 2. Materials and Methods

### 2.1. Materials and Apparatus

Polyethylene naphthalate (PEN, supplied by Goodfellow, Ltd., Huntington, UK) with an area of 1 × 1 cm^2^ and a thickness of 50 µm was used in our experiments. Before modification, each sample was cleaned to remove mechanical impurities by a stream of nitrogen. The modification was carried out using a pulsed KrF laser @248 nm (COMPexPro 50 F, Coherent, Inc., Santa Clara, CA, USA) equipped with a linear polariser (UV-grade fused silica prism, model PBSO 248-100). To form LIPSS, the processing parameters were as follows: 6000 pulses, pulse duration 20–40 ns, frequency 10 Hz, aperture with an area of 10 × 5 mm^2^, laser fluence 5.9 mJ∙cm^−2^. The incidence angle of the laser beam (22.5°) was selected based on the optimal surface morphology derived from ref. [39].

The deposition of Cu nanowires (CuNWs) on the rippled PEN was realized by vacuum evaporation under specific conditions according to our previous study [24]. However, the copper deposition in this work was accomplished only from one side of the ripple structure. We used copper sheets (0.5 g, 99.99% purity, Safina a.s., Vestec, Czech Republic) that were placed in a resistively heated tungsten crucible. During deposition, the thickness of the nanowires was controlled by the dynamic weighing method using quartz crystal oscillations to achieve a thickness of 10 nm. To verify the thickness of the nanowires, the scratch method was used [40] and the thickness of Cu was measured at five different sample positions by atomic force microscopy (AFM, VEECO CP II, Veeco Instruments Inc., New York, NY, USA). Thickness variation did not exceed 5%. After deposition, the selected samples (marked as Cu/LIPSS/PEN*) were left for 4 months under laboratory conditions (T = 25 °C, ambient atmosphere) to record the time dependence of the selected parameters.

### 2.2. Analytical Methods

All samples were analysed immediately after their preparation. In the case of Cu nanowires, the time dependence of the surface morphology and chemical composition was monitored.

The surface morphology of the pristine polymer (PEN), polymer with LIPSS structure (LIPSS/PEN), and Cu nanowires supported on a rippled PEN (Cu/LIPSS/PEN) was acquired by atomic force microscopy (AFM VEECO CP II, Veeco Instruments Inc., New York, NY, USA) in ‘tapping mode’. Measurements were conducted with an oxide-sharpened P-doped silica probe RTESPA-CP (spring constant of 0.9 N·m^−1^, near-resonant frequency 300 kHz, Veeco Instruments Inc., New York, NY, USA). Surface scanning was performed with a frequency of 0.5 Hz in four different sample areas to obtain representative data. Surface characteristics derived from the AFM measurements, such as average surface roughness (*R*_a_), periodicity (*Λ*), and height (*h*) of the created nanostructures, were evaluated using the NanoScope Analysis v1.4 software (Bruker, Billerica, MA, USA). The deviations of the measured parameters did not exceed 5%.

The sample morphology was investigated using scanning electron microscopy (SEM) with an FEG electron gun (SEM, LYRA3GMU, TESCAN, Brno, Czech Republic) at an acceleration voltage of 20 kV. Elemental analysis was performed using an energy dispersive spectroscopy (EDS) analyser with an 80 mm^2^ SDD detector (X-MaxN, Oxford instruments, Abingdon, UK) and AZtecEnergy v3.1 software (Oxford instruments).

Angle-resolved X-ray photoelectron spectroscopy (AR-XPS) was used for the analysis of the chemical composition of pristine PEN, PEN with LIPSS structure, and prepared nanowired composites. For analysis, an ESCAProbeP spectrometer (Omicron nanotechnology GmbH, Taunusstein, Germany) with a monochromatic X-ray beam source with an energy of 1486.7 eV was used. The samples were analysed stepwise with a step of 0.05 eV at angles of 90° and ±9° with respect to the orientation of the sample. The atomic concentration of elements (at. %) was evaluated with the CasaXPS v2.3 software (Casa Software Ltd., Teignmouth, UK).

The absorption spectra of the samples were measured by ultraviolet-visible spectroscopy (UV-Vis) using a Lambda 25 spectrometer (Perkin-Elmer, Inc., Waltham, MA, USA). The measured spectra were processed by Perkin Elmer UV WinLab v4.2 software (Perkin-Elmer, Inc., Waltham, MA, USA). Measurement parameters were set as follows: a scanning range of 400–900 nm, a scanning rate of 240 nm∙min^−1^. The absorption spectra of the samples with CuNWs were measured on the same device using the WP25L-UB polariser (250–4000 nm, Thorlabs, Newton, NJ, USA) perpendicularly to the orientation of the nanowires.

The hydrophobic/hydrophilic properties of the pristine, LIPSS/PEN and Cu/LIPSS/PEN samples were determined by measuring the contact angles using KRÜSS DSA 100 goniometer (KRÜSS, Hamburg, Germany) by a sessile drop method. Ten drops of 2 µL of distilled water were applied to the sample surface by an automatic pipette. Then, the contact angle was calculated using a three-point method, and the arithmetical means and standard deviations were determined from the obtained data. The contact angle values were evaluated using the KRÜSS Advance v2.0 software.

## 3. Results and Discussion

The surface morphology and chemical composition of the sample surface were studied with AFM, SEM, SEM-EDS, and XPS. Surface morphology, together with related surface parameters such as surface roughness (*R*_a_), periodicity (*Λ*), and height (*h*) are shown in Figure 1 and Table 1, respectively. It is obvious from Figure 1 that the pristine PEN showed a flat surface with a surface roughness of 4.27 nm. After laser modification, the polymer surface was transformed into coherent ripple structures (LIPSS) with pronounced surface roughness [22,24,39]. This dramatic change was related to the LIPSS formation; a complex process based on the redistribution of matter due to refraction and interference of incident radiation at the interface with different refractive indexes [14].

A significant increase in *R*_a_ and *h* was observed in the cases of the Cu/LIPSS/PEN and Cu/LIPSS/PEN* samples. This increase was related to a change in surface chemistry, which led to a preferential interaction of copper with the oxygen-rich surface. In this case, the nucleation occurred on the ridges of the ripples, and the subsequent growth of nanowires resulted in nanostructures higher than those of LIPSS/PEN. Some similarities can be observed in a study by Siegel et al. [41], where the deposited gold nanolayer was presented exclusively on the ridges, so the condensed material was formed into nanowires during sputtering. Although the required thickness of the deposited Cu was 10 nm, the height of the CuNWs derived from AFM was roughly 40 nm. The metal layer was probably enriched with oxidised copper because of the interaction of the residual oxygen with the flow of the evaporated copper together with post-deposition oxidation of the metal structures.

The most striking result that emerges from Figure 1 is the presence of clusters on the entire surface of the individual wires. We assumed that the formation of clusters was due to an oxidation process in the deposited copper layer, which is further supported by SEM and EDS analyses.

SEM analysis was performed to reveal the metal–polymer interface and the surface morphology of the Cu/LIPSS/PEN* samples due to the pronouncement of material contrast (metal/polymer). We simultaneously recorded the signal of secondary electrons (SE) and back-scattered electrons (BSE) from the same area of the sample (Figure 2). The image of the secondary electrons, which are sensitive to surface roughness, corresponds to a periodicity of 260 nm. From Figure 2, the interface of copper-PEN (highlighted with arrows) and the formed clusters is visible. The back-scattered electrons image shows differences in the atomic number, which means that the higher the image intensity, the higher the atomic number. Thus, the most intense parts of the image correspond to copper in the Cu/PEN system, the Cu being deposited mainly on one side of the ripples near the ripple top, which is in accordance with the vacuum evaporation mechanism. This phenomenon is known as the shadow effect [42].

To confirm the Cu oxidation, we acquired EDS maps of the Cu/LIPSS/PEN and Cu/LIPSS/PEN* samples. Direct imaging of EDS counting maps is ambiguous as the rippled structure of the surface caused a variation in the EDS counts from point to point. Specifically, we observed a more intensive EDS signal for all the detected elements (Cu, O, and C) at the ripple tops. To eliminate the roughness effect, we performed pixel-by-pixel spectrum deconvolution to obtain maps showing the distribution of the atomic percent, since the ratio between the Cu, O, and C peaks is independent of the total EDS signal intensity. Then, to detect a change in the oxidation state, we calculated the atomic ratio of Cu/O and presented the ratio on a false colour map (Figure 3). The Cu/O values are smaller than those expected from the standard stoichiometry of copper oxides. The reason is that the ratio is affected by the background oxygen signal from PEN. Therefore, the samples could only be compared with each other. We see the precise localization of a higher Cu/O ratio on top of the ripples in the Cu/LIPSS/PEN samples, whereas the Cu/LIPSS/PEN* samples show the delocalization of the Cu/O ratio. Furthermore, the mean values calculated over all pixels in the corresponding maps show a decrease in the Cu/O ratio of about 20% after material ageing, which again indicates the oxidation of the Cu nanowires. This finding is in accordance with the selective oxidation of CuNWs that is captured on AFM images in the form of distinctive clusters (see Figure 1, Cu/LIPSS/PEN*).

The elemental composition of the surface of the pristine, LIPSS/PEN, Cu/LIPSS/PEN, and Cu/LIPSS/PEN* samples was determined by angle-resolved XPS. Concentrations of oxygen (O), carbon (C), and copper (Cu) in at. % are shown in Figure 4. The atomic concentration of Cu for samples with CuNWs was higher at an electron take-off angle of 90° compared to angles of ±9°. Some similarities with our results can be found in a study by Kaimlová et al. [39], where the concentration of deposited Ag on laser-modified PEN was the highest at 90°. Moreover, analytical information on the perpendicular direction is determined from a larger volume of interaction compared to measurements at smaller angles, leading to different concentrations of elements [43]. To clarify the arrangement of CuNWs, measurements at electron take-off angles of +9° and −9° were also accomplished. From Figure 4, it is clear that the Cu content in the Cu/LIPSS/PEN and Cu/LIPSS/PEN* samples was higher at an angle of +9°, which confirms the condensation of CuNWs mainly on one side of the ripples associated with the shadow effect. The concentration difference is not significant for the old samples due to the oxidation of copper and the presence of clusters on the entire PEN surface. The atomic concentration of O slightly increased after PEN irradiation with a KrF laser, leading to a rearrangement of the atoms and partial oxidation of carbon-based groups [44]. Because we were interested in the oxidation of CuNWs, we focused on the changes in O concentration for the Cu/LIPSS/PEN and Cu/LIPSS/PEN* samples. After 4 months, copper was dramatically oxidized, as the increase in O was about 30 at. % compared to the as-prepared samples.

The optical properties of the samples were analysed by UV-Vis spectroscopy. In Figure 5, the absorption spectra of the pristine, LIPSS/PEN, and Cu/LIPSS/PEN samples are shown. The absorbances of PEN and laser-treated PEN were similar in the range of visible radiation. Typically, polymers with an aromatic core interact with UV radiation due to the presence of conjugated double bonds [45,46]. The absorbance of LIPSS/PEN slightly increases with decreasing wavelength due to the incorporation of new conjugated double bonds during laser modification.

Significantly different absorption was found in the LIPSS/PEN and Cu/LIPSS/PEN samples. After the deposition of the metallic structures, the absorbance increased considerably. This increase can be explained by the oscillation of delocalized electrons as a result of their interaction with electromagnetic radiation. This phenomenon is known as surface plasmon resonance (SPR), which is expressed in the UV-Vis spectrum as a significant SPR peak. The absorption maximum measured for the Cu nanowires was at a wavelength of 594 nm. Similar results were obtained for Cu nanowires prepared by electrodeposition in an experiment by Lotey et al. [47] and for Cu nanoparticles synthesized by chemical reduction studied by Julkarnain et al. [48].

The surface wettability of the samples was examined by measuring the contact angles (CA) for the pristine, LIPSS/PEN, and Cu/LIPSS/PEN samples. In Figure 6, images of a distilled water drop with appropriate CAs are shown for all types of samples. In the case of PEN and LIPSS/PEN, no differences in CA were detected within the error. A marked increase in CA was found for the deposited samples, which reached (108.9 ± 1.5)°. This increase is likely caused by the presence of CuNWs. However, a hydrophobic surface could be presented as a result of the change in surface morphology. The values of surface characteristics, such as height and periodicity (obtained by AFM), probably influence the value of CA measured immediately after the application of the water droplet. The water droplet was in the Cassie-Baxter state, so considerably higher nanostructures with Cu nanowires (75 nm) provided grooves with trapped air [49,50]. Overall, these results suggest that the deposition of copper nanowires led to nanocomposites with a hydrophobic character.

In summary, the Cu-PEN composite prepared in this way could potentially be used in biomedical applications. The oxidation of CuNWs seems to be problematic for this field of study; nonetheless, Cu oxides are also referred to as an antibacterial agent [51,52] or even an improvement in biocompatibility [53,54]. For this reason, in our further experiments we intend to focus on the biological responses of such engineered surfaces. It is important to point out that the usage of CuNWs as surface coatings must be thoroughly discussed depending on the application, and its oxidation should be further investigated in different biological environments.

## 4. Conclusions

In this study, we combined surface treatment and Cu nanowire deposition to enhance the properties of PEN foil. Laser surface treatment by linearly polarized light from the KrF excimer laser and vacuum evaporation of the metal provided uniform copper nanowires with an effective thickness of 40 nm. The AFM images showed an increase in surface roughness after copper deposition. AR-XPS revealed that CuNWs were deposited on one side of the ripples near the ridge. The surface chemistry was influenced by laser treatment, which was supported by photoelectron spectra. After 4 months, the roughness slightly decreased, and the Cu oxides were homogeneously distributed within the CuNWs, which was detected by EDS and indirectly observed also by AFM and SEM. The prepared nanocomposites exhibited a hydrophobic character due to the presence of Cu. In general, our approach may be used advantageously in the production of polymer-supported nanostructures with specific surface characteristics, which could be used in a wide range of applications. However, attention must be paid to the oxidation of the structures, which is quite intense and even changes the morphology of the composites.

## Figures and Tables

**Figure 1 nanomaterials-12-01220-f001:**
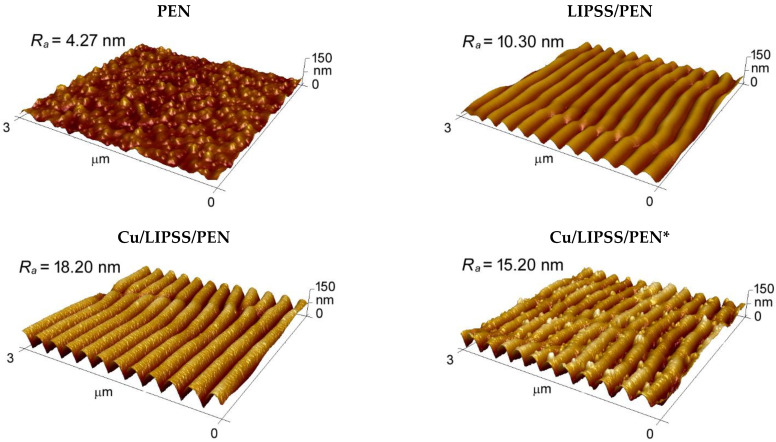
3D AFM images of pristine, LIPSS/PEN, Cu/LIPSS/PEN, and Cu/LIPSS/PEN* samples with the corresponding values of surface roughness (*R*_a_).

**Figure 2 nanomaterials-12-01220-f002:**
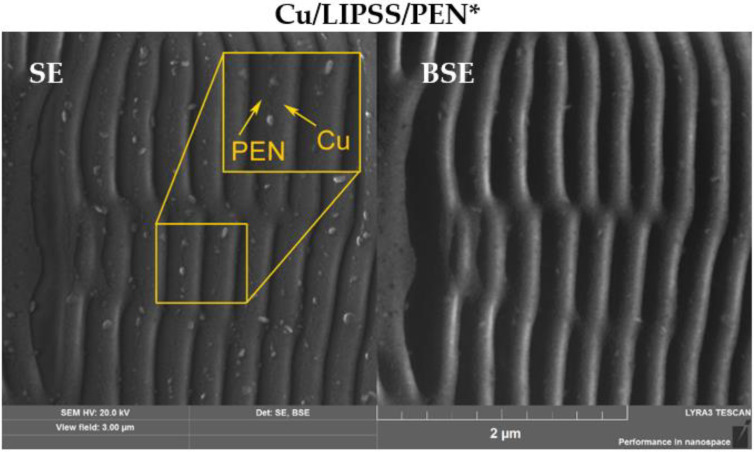
SEM micrographs of Cu nanowires supported on PEN with a highlighted Cu-PEN interface.

**Figure 3 nanomaterials-12-01220-f003:**
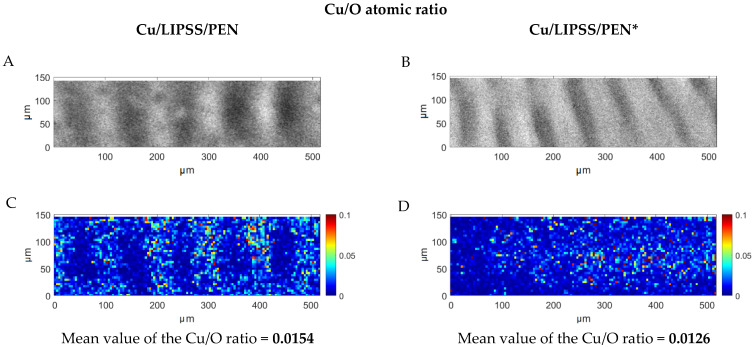
SEM images (**A**,**B**) and the corresponding EDS maps (**C**,**D**) showing the Cu/O ratio at. %. The mean values were calculated for each map.

**Figure 4 nanomaterials-12-01220-f004:**
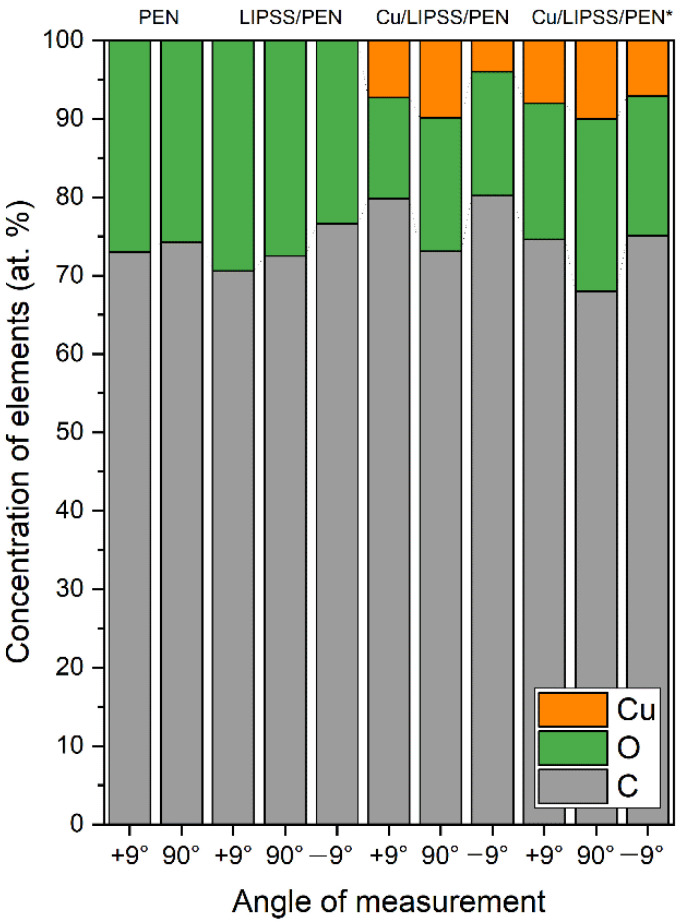
Atomic concentrations of elements (Cu, C, O) derived from XPS analysis for pristine PEN, LIPSS/PEN, and Cu/LIPSS/PEN immediately after Cu deposition (Cu/LIPSS/PEN) and 4 months after Cu deposition (Cu/LIPSS/PEN*).

**Figure 5 nanomaterials-12-01220-f005:**
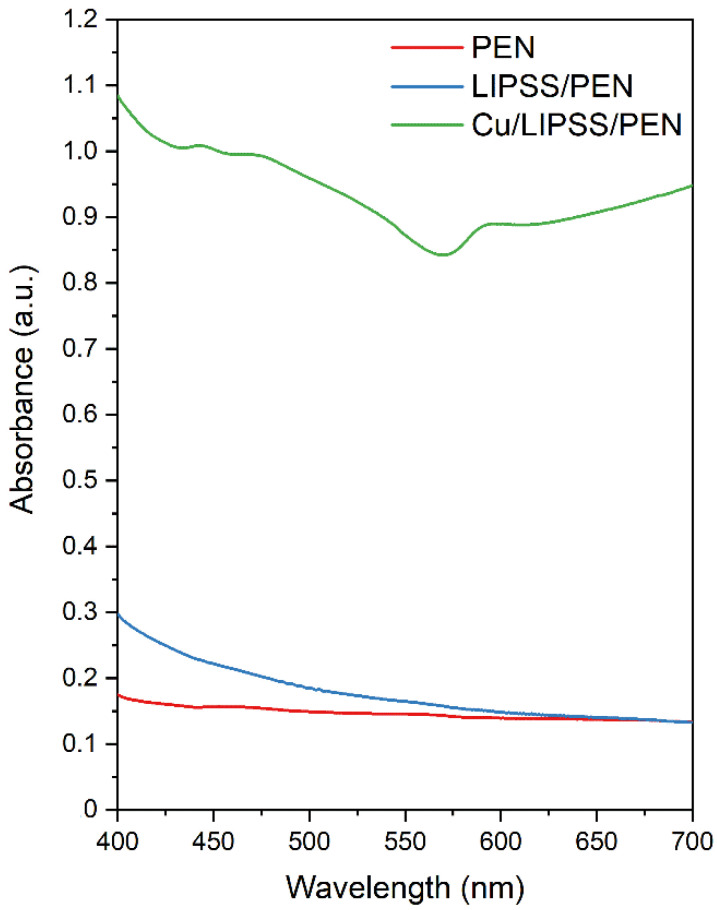
Absorption spectra of pristine, LIPSS/PEN, and Cu/LIPSS/PEN.

**Figure 6 nanomaterials-12-01220-f006:**
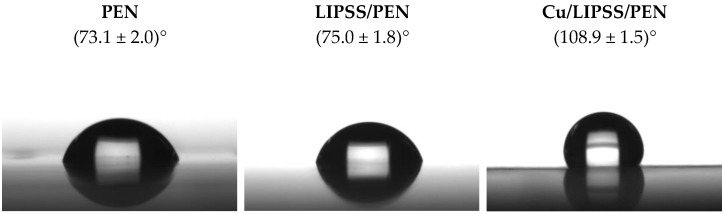
Images of water droplets on the surfaces of PEN, LIPSS/PEN, and Cu/LIPSS/PEN with corresponding contact angles.

**Table 1 nanomaterials-12-01220-t001:** Surface characteristics: average surface roughness (*R*_a_), periodicity (*Λ*), and height (*h*) derived from AFM.

Sample	*R*_a_ (nm)	*Λ* (nm)	*h* (nm)
PEN	4.27	-	-
LIPSS/PEN	10.30	262.80	34.70
Cu/LIPSS/PEN	18.20	259.20	73.60
Cu/LIPSS/PEN*	15.20	260.20	75.90

## Data Availability

The data presented in this study are available on request from the corresponding author.

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
