# Peer review of "Engineered Cu-PEN Composites at the Nanoscale: Preparation and Characterisation"

_nanomaterials, 2022, doi:10.3390/nano12071220_

Round 1

Reviewer 1 Report

The work presents preparation of Cu-nanowires on the nanostructured PEN substrate. Generally, the paper presents the study and the results comprehensibly. However, the big concern is its novelty. The nanostructuring by KrF eximer laser of PEN substrate as well as formation of metal (Ag, Au) nanowires on it were presented in other works of these authors. Therefore, the focus of this work should have been the useful properties related with specific applications of the prepared Cu nanowires/PEN structures. This is not the case and there is only a short presentation of hydrophobic character of these structures and the very general statement that it could be used in biomedical applications. No tests related to the possible biomedical applications (e.g. antibacterial, cytotoxicity, etc.) are presented. Moreover, the reported oxidation of structures with time is not in favor of such applications.

Therefore, the reviewer believes that the work does not contribute sufficient novelty and should be enriched with additional application related investigations.

Author Response

Sincerely yours

Jana Pryjmaková

Reviewer 2 Report

The manuscript "Engineered Cu-PEN composites at the nanoscale: Preparation and Characterisation" reports an interesting method for the preparation of nanostructured composite polymer/copper surfaces. After the polymer nanostructuring with laser and Cu deposition, the authors carried out an extensive morphological, chemical, optical and interfacial characterization. Such characterization demonstrated that Cu deposits preferentially on one side of the nanostructured ripples leading, with respect to the nanostructured polymer film, to an increase of the nanostructure height. Moreover, SEM and XPS were employed to demonstrate the partial oxidation of the deposited copper and the increase of such oxidation with sample aging. However, this characterization needs, in my opinion, one major revision: the author should report the XPS data to show the evolution of the copper/oxygen ratio and, more importantly, observe and discuss if any chemical shift or change of the shape of the copper peak is observed with aging. Given the increased oxidation reported by the authors, this should be the case and the only report of the XPS-derived elemental analysis data is not sufficient.

Then, the authors performed an optical characterization, which showed the onset of surface plasmon resonance absorption due to Cu deposition, and contact angle measurement showing the hydrophobicity increase after Cu deposition. As the authors correctly state, the hydrophobicity increase may be due to the increased sample roughness but its origin could also be related to the deposition of hydrophobic metallic copper. I believe that more information about it could be obtained by the measurement of the contact angle of the aged sample, which is not reported yet. Thus, the authors are encouraged to show and discuss the contact angle of the aged sample.

Two more minor issues:

_ The name Cu/LIPSS/PEN*, which I assume refers to the aged sample, is never properly introduced. I believe its initial explanation could help the readership

_ Page 1, lines 42-43 "The wide variety of surface modification techniques, such as physical and chemical deposition of the thin layer, ion implantation, Langmuir-Blodgett coating, or plasma and laser treatment..." The authors are encouraged to provide bibliographic references to all the polymer deposition methods here cited.

Author Response

Sincerely yours

Jana Pryjmaková

Round 2

Reviewer 1 Report

The authors provided explanation and answered my concern about the novelty of the work. Generally, I’m accepting it and could consider the present work as a starting point for further dedicated investigations of the properties of Cu:PEN nanostructures related to their possible application. Some relevant improvements of the text have also been made.

Reviewer 2 Report

The authors have addressed my concerns. I feel now the article can be published